# Data Mining as a Tool to Infer Chicken Carcass and Meat Cut Quality from Autochthonous Genotypes

**DOI:** 10.3390/ani12192702

**Published:** 2022-10-08

**Authors:** Antonio González Ariza, Francisco Javier Navas González, José Manuel León Jurado, Ander Arando Arbulu, Juan Vicente Delgado Bermejo, María Esperanza Camacho Vallejo

**Affiliations:** 1Department of Genetics, Faculty of Veterinary Sciences, University of Córdoba, 14071 Córdoba, Spain; 2Agropecuary Provincial Centre, Diputación Provincial de Córdoba, 14071 Córdoba, Spain; 3Institute of Agricultural Research and Training (IFAPA), Alameda del Obispo, 14004 Córdoba, Spain

**Keywords:** biodiversity, sustainability, local genetic resources, native breeds, chicken meat, chemical characterization, physical traits, meat cuts

## Abstract

**Simple Summary:**

The present study is a meta-analysis of ninety-one research documents dealing with carcass quality characterization in autochthonous chicken genotypes. Documents were published between 2002 and 2021. Data mining methods were used to determine which variables should be considered or otherwise discarded from comprehensive carcass quality differential models to improve the study’s efficiency and accuracy. Even if the impact on carcass quality of certain variables such as chicken sex, meat firmness, chewiness, L* meat 72 h post-mortem, a* meat 72 h post-mortem, b* meat 72 h post-mortem, and pH 72 h post-mortem could be presumed, these should not be considered if strongly related variables are simultaneously considered too, to prevent redundancy problems. In contrast, carcass/cut weight, pH, carcass yield, slaughter age, protein, cold weight, and L* meat must be regarded strictly due to their high potential to explain differences and correctly classify carcass cuts across chicken genotypes. The standardization of characterization methods of minority populations (with limited censuses and lacking population structure, but well-adapted to alternative systems) enhances the possibility of success of the implementation of sustainable conservation strategies through the dissemination of knowledge on local breeds and the competitivization of their distinctive products within specific market niches.

**Abstract:**

The present research aims to develop a carcass quality characterization methodology for minority chicken populations. The clustering patterns described across local chicken genotypes by the meat cuts from the carcass were evaluated via a comprehensive meta-analysis of ninety-one research documents published over the last 20 years. These documents characterized the meat quality of native chicken breeds. After the evaluation of their contents, thirty-nine variables were identified. Variables were sorted into eight clusters as follows; weight-related traits, water-holding capacity, colour-related traits, histological properties, texture-related traits, pH, content of flavour-related nucleotides, and gross nutrients. Multicollinearity analyses (VIF ≤ 5) were run to discard redundancies. Chicken sex, firmness, chewiness, L* meat 72 h post-mortem, a* meat 72 h post-mortem, b* meat 72 h post-mortem, and pH 72 h post-mortem were deemed redundant and discarded from the study. Data-mining chi-squared automatic interaction detection (CHAID)-based algorithms were used to develop a decision-tree-validated tool. Certain variables such as carcass/cut weight, pH, carcass yield, slaughter age, protein, cold weight, and L* meat reported a high explanatory potential. These outcomes act as a reference guide to be followed when designing studies of carcass quality-related traits in local native breeds and market commercialization strategies.

## 1. Introduction

Poultry production has rapidly developed worldwide [1]. Contextually, big companies are responsible for the major management of (optimized housing and feeding conditions have considerably shortened the rearing period of fast-growing chickens) and genetic improvements made to intensive farming systems in response to the current dependence on and demand for chicken meat [2].

While intensive poultry farming almost exclusively relies upon high-yielding commercial hybrid ‘only meat’ strains [3] to provide a large amount of meat for the human population on a large scale [4], indigenous chicken populations still significantly contribute to local economies, especially low-income rural livelihoods, across Asia, Africa, South America, and the South Pacific [5,6,7]. As a result, the replacement and hybridization of native breeds with these exotic strains, which may internationally be more commercially competitive, drastically threatens the genetic diversity of worldwide poultry populations [8].

In this regard, although the current promotion of specialized layer or meat producer genotypes on chicken farms has produced a displacement and marginalization of dual-purpose systems (supplanting native chicken breeds) [9], problems linked to meat maturity and its technological and sensory quality [10] may arise. To counteract such problems, a growing demand for poultry products from alternative production systems has brought about the opportunity to increase the importance of raising autochthonous breeds.

According to the DAD-IS (Domestic Animal Diversity Information System) FAO database [11], only 9.18% of local breeds are actually not at risk (Figure 1). Local genotypes may be the source of enhanced distinctive products, and may play a rather pivotal role in meat quality over quantity. In this regard, poultry meat quality can be understood in various ways, ranging from poultry meat nutritive value to sensory traits, among others [12].

The importance of such local genotypes relies on the fact that they thrive on elements of organic and free-range farming due to the suitability of their nature to adapt to their origin area [13]. This has been confirmed by the international experience of several countries, where slow-growing native chicken breeds have been able to provide good-quality meat, at a reasonable price, which is the main rationale behind the increasing demand for distinctive products [14].

On the one hand, nutritional quality comprises the content of macro- and micronutrients, unsaturated fatty acids, high-value protein, cholesterol, and other biologically active compounds. On the other hand, organoleptic quality considers sensory-related desirable traits such as meat flavour, aroma, and colour as essential traits to monitor [15]. It is the simultaneous evaluation of both which may determine the right market niche and target consumers for which each meat cut or type may be aimed [16].

Last but not the least, meat quality is subject to trends, and current consumer tastes are characterized by the challenge of obtaining low-fat meat products that preserve all the tenderness, juiciness, and good flavour and aroma of high-fat meat products [17,18]. Additionally, meat quality traits are influenced by factors of a very different nature, such as the slaughter age of birds, the feed provided to them [19], or genetic factors inherent to the genotype which the meat cuts derive from. Hence, making the appropriate choice of a specific chicken breed/variety or commercial hybrid is necessary if our aim is to maximize the expected commercial outcomes.

For these reasons, the first objective of the present study is to determine the differential clustering patterns described by the carcass and meat quality-related traits defining the cuts of meat of worldwide local chicken breeds. Second, the benefits that derive from the use of data mining are verified through the development of a functional tool to quantify the similarities and dissimilarities across carcass cuts derived from autochthonous chicken genotypes whose product quality or quantity analysis has been previously scientifically studied. The outcomes of the present study will help to tailor specific solutions to fulfill the needs of certain market niches based upon those meat cuts derived from alternative poultry farming and locally adapted breeds worldwide. Moreover, the tool that has been developed in this study may help plan the methodology for future research involving minority populations of chickens when seeking to evaluate meat quality traits in particular.

## 2. Materials and Methods

### 2.1. Systematic Review Approach Decision

The approach followed in the present systematic review has been reported to be an efficient tool in the scope of animal science specific topics [20,21,22]. PRISMA guidelines were discarded, given that PRISMA criteria for systematic reviews were developed in the scope of healthcare research; hence, this does not fit the diverse range and nature of the documents in which the information in regard to local breeds is made available to the public [23]. This has been supported by studies such as that by Tam et al. [24], who suggested that the adherence level of certain journals to the PRISMA statement does not significantly change whether they endorse or recommend such a guideline. Furthermore, other authors such as Haddaway et al. [25] report the limited applicability of PRISMA guidelines for reviews in conservation and environmental management.

### 2.2. Data Collection

Our data collection methodology followed the premises described in previous studies [20,21,22]. Two platforms (www.google.scholar.es and www.sciencedirect.com; accessed on 27 May 2022) were used for the document search [26]. The possibility to extract data from repositories was an applied inclusion criterion. In this regard, although the aforementioned repositories permit data extraction for further process and assessment, other repositories such as www.webofscience.com/wos/woscc/basic-search and www.ncbi.nlm.gov/pubmed/ (accessed on 27 May 2022) do not. Thus, this fact prompted their exclusion as information sources. Non-open access full manuscripts were accessed via the University of Córdoba library service. The document search was performed using the subsequent keyword list h: carcass or meat quality/characterization followed each one with the words local/native/indigenous/autochthonous poultry or chicken breed, or any related term in their semantic fields [20,27]. After document collection, our study database comprehensively contained documents published from 2002 to 2021. The document search was completed by 31 December 2021 to ensure the document database comprised all documents published during 2021.

As an inclusion criterion, only those research documents which involved breeds cataloged as native in the DAD-IS database were considered for statistical analyses [28]. As a result of this selection process, 91 publications, published in English and Spanish languages, were selected to evaluate the quality of different meat pieces from different local chicken genotypes. Traits evaluated in the documents were sorted into clusters, as shown in Table 1. Unit conversion was carried out to standardize the information reported in the different documents so as to be able to quantify the quality of the different carcass pieces across all the breeds that were studied. The most widely used unit across the documents considered for each particular variable was chosen as a reference for unit conversion.

The information present on each document was sorted into the different study observations depending on the following factors: breed, sex, sex status, slaughtering age, and meat cuts. In regard to sex/sex status, the possibilities (levels) considered were female, male, both (when females and males were used in the documents without being reported separately), capon, and poulard.

Thirty-five different meat cuts or carcass components were studied as follows: abdominal fat, back, blood, breast, caeca, carcass (whole carcass), carcass remainder, comb, drumstick, feathers, giblet, gizzard, head, heart, intestine, liver, lungs, neck, ovary, pancreas, pelvis, proventriculus, rear, ribs, shanks, skeletal, skin, spleen, testes, thighs, thymus, trunk, viscera (whole viscera), wattles, and wings.

On the whole, 39 variables were included in the statistical analyses: sex (sex and sex status), slaughtering age, carcass/piece weight, carcass yield, cold weight, drip loss, water-holding capacity, cooking loss, L* meat, a* meat, b* meat, L* meat 72 h post-mortem, a* meat 72 h post-mortem, b* meat 72 h post-mortem, L* skin, a* skin, b* skin, muscle fiber density, muscle fiber diameter, drip loss, water-holding capacity, cooking loss, firmness, total work, shear force, hardness, springiness, cohesiveness, gumminess, chewiness, pH, pH 24 h post-mortem, pH 72 h post-mortem, IMP, AMP, inosine, moisture, protein, fat, ash, collagen, and cholesterol.

All techniques and methodologies followed in the different research documents to collect the measurements of each particular explanatory variable were standardized and described in the research procedures present in each document. For this reason, the rationale behind the present research was not to infer about the methods provided, as reported in the literature, when standardized laboratory techniques were used; even if empirical differences may have been detected at first sight, these differences were statistically nonsignificant [128,129].

### 2.3. Data Analysis

#### 2.3.1. Multicollinearity Prevention: Preliminary Testing

Before performing the statistical analyses per se, a multicollinearity analysis was run to discard potential strong linear relationships across explanatory variables and ensure data independence. In this way, before data manipulation, redundancy problems can be detected, which limits the effects of data noise and reduces the error term of discriminant models. The multicollinearity preliminary test helps to identify unnecessary variables which should be excluded, preventing the overinflation of variance explanatory potential and type II error increase [130].

The variance inflation factor (VIF) was used to determine the occurrence of multicollinearity issues. The literature reports a recommended maximum VIF value of 5 [131]. On the other hand, tolerance (1 − R^2^) concerns the amount of variability in a certain independent variable which is not explained by the rest of the dependent variables considered (tolerance > 0.20) [132].

The multicollinearity statistics routine of the describing data package of XLSTAT software (Addinsoft Pearson Edition 2021, Addinsoft, Paris, France) was used. The following formula was used to calculate the VIF:VIF = 1/(1 − R^2^),(1)
where R^2^ is the coefficient of determination of the regression equation.

#### 2.3.2. Data-Mining Chi-Squared Automatic Interaction Detection (CHAID) Decision Tree: Splitting, Pruning and Building

The CHAID decision tree was used to classify, predict, interpret, and develop discrete categorized data tool inference. The tree routine of the Analyzing Data package of the XLSTAT software (Addinsoft Pearson Edition 2021, Addinsoft, Paris, France) was used.

In the decision tree, each internal node was built around an input variable (meat or carcass quality traits) when a significance split criterion of the chi-square test (*p* < 0.05) in the so-called pre-pruning process was met.

Pre- or post-pruning methods prevent the oversizing of trees to avoid failures by seeking the addition of traits (branches) that significantly add to the overall fit [133]. Nodes that did not significantly contribute to the global prediction were discarded. After the process, the tree obtained exhaustively represents the significant relationships across the levels of the dependent variable. Additionally, CHAID is used to penalize model complexity through an adjustment of Bonferroni inequality by significance levels.

Consecutive chi-squared tests are performed during the tree-building configuration process [134]. While branches represent the test results (in a number of two or more), the leaf nodes (or terminal nodes) represent the category levels of the target variables (the piece of carcass). Classification decisions are made at each node from the first root node placed at the top of the tree. Each data record is explored along the tree until it reaches a terminal node or leaf. The correlation matrix obtained from the development of the data mining analysis was graphically depicted through the use of the web server Heatmapper (www.heatmapper.ca; accessed on 30 June 2022) [135].

#### 2.3.3. CHAID Decision Tree Cross-Validation

Ten-fold cross-validation was performed to ensure that the set of predictors considered significantly explains the differences across dependent variable groups to validate the outcomes of the CHAID decision tree. All sample records of the training sample and the study data were used to perform the ten-fold cross-validation [133].

For ten-fold cross-validation, we created 10 random subsets of the original data, setting one fold aside which was used as a test set. Afterwards, we built a tree for the remaining folds (10 − 1), and evaluated the tree, comparing it against the test fold. Then, we built one tree with the remaining 90% of the cases for each of the 10 subsets (subsamples). The 10% subset was treated as a test sample (subset). For a 10-fold validation, each of the 10% folds (mutually exclusive and summing up to the total observations in the sample) at once serve as a test sample and as part of the learning sample 9 times. Cross-validation compares the differences between prediction errors for a tree applied to a new potential sample (resubstitution error rate) and a training sample (cross-validation error rate).

The ‘complexity parameter’ (cp) was used to perform the cross-validation of the decision. The complexity parameter (cp) controls the size of the decision tree and helps to select the optimal tree size. When adding another variable to the decision tree from the current (lowest) node implies a statistical cost or increases the complexity of the discriminant model above the value of cp, then tree building stops.

The resubstitution or replacement rate refers to the proportion of misclassified original observations across the various subsets of the original tree. The resubstitution rate decreases as tree depth increases. The lowest resubstitution/replacement error rates are yielded by the largest tree. Notwithstanding, selecting trees reporting the lowest resubstitution rate may not be the best choice due to the potential bias derived from redundant variable inclusion. Large trees add random variation to the predictions given they overfit outliers. 

Ten-fold cross-validation was used to obtain a cross-validation error rate (risk). The cross-validation risk is an averaging of the risks across the 10 test samples (folds) This process was repeated for each fold, evaluating an estimate of such error. The sum of the error in the 10 portions represented the cross-validation error rate. Finally, the tree that produced the lowest cross-validation error rate and, therefore, presented the best fit, was selected. The best tree can be defined as the tree closest to the minimum. Hence, it can be used to determine the accuracy of the discriminant model for data prediction. Contextually, Albayrak [136] reports that the optimal tree depth can be identified as the shallowest tree whose cross-validation risk does not exceed the risk of the minimum cross-validation risk tree, plus one standard error of this tree’s cross-validation risk. This means that the resubstitution error rate and the cross-validated error rate must be compared to choose the optimal tree as a counteracting measure of the bias derived from the overfitting of outliers.

## 3. Results

### 3.1. Multicollinearity Prevention: Preliminary Testing

A summary of values for VIF and tolerance is reported in Table 2. Variables whose VIF values were ≥ 5 were discarded from further analyses. Thus, the traits removed for the following statistical analyses were sex, L* meat 72 h post-mortem, a* meat 72 h post-mortem, b* meat 72 h post-mortem, firmness, chewiness, and pH 72 h post-mortem.

### 3.2. Data-Mining Chi-Squared Automatic Interaction Detection (CHAID) Decision Tree: Splitting, Pruning, and Building

The data-mining CHAID decision tree obtained from the chi-square dissimilarity matrix built in this study is represented in Appendix A. A correlation matrix across carcass and meat quality-related traits were computed and are graphically represented in Figure 2. The chi-square-based branch and node distribution suggested that observations significantly (*p* < 0.05) differed across carcass meat cuts. Appendix A report the frequency distribution of the presence of each cut across the range of levels for the particular carcass or meat quality trait represented by the different nodes within the tree structure.

The first three branches of the CHAID decision tree are summarized in Figure 3. Five groups were depicted in the first classification (first node: 0–14.46; second node: 14.46–47.33; third node: 47.33–69.76; fourth node: 69.76–80.76; fifth node: 80.76–87.14), derived from the representativity of carcass yield for the different pieces. After this, observations were sorted into subnodes originating depending on the values for pH and carcass/piece weight in each meat cut.

### 3.3. CHAID Decision Tree Cross-Validation

Finally, the validity and robustness of the results were cross-validated and the number of erroneously classified observations for each piece was computed. As reported in Table 3, the risk estimates (≈0.600) and standard errors (0.013) of the model applying the cross-validation test did not differ from the results of the model without the cross-validation test. Hence, the stability of the model was guaranteed.

## 4. Discussion

The present research develops an updated evaluation of international research studies focusing on carcass characterization in autochthonous chicken breeds worldwide. The imbalance between the economic resources allocated to native genotypes when compared with commercial hybrid strains produces a gap in the knowledge, visualization, and impact that such local genotypes will eventually have in the research community and by extension in society [137].

Recent decades have been characterized by a progressively reducing trend in genetic diversity. Such a reduction has not only affected diversity across chicken genotypes, but also within-genotype diversity. Such a lack of diversity compromises one of the main needs that poultry production seeks to fulfil: the provision of a sufficiently diverse genetic background so as to face the adaptation to climate change and meet consumer preferences, as well as current and forthcoming market demands [138]. In this regard, to ensure such an objective is attained, breeds’ long-term survival cannot be left aside, and for this the knowledge on such breeds must be deepened. The characterization of local resources and of the products which derive from them is, therefore, critical.

Local breeds have proven to be sources for products which are well appreciated in specialized market niches, although their lower productivity in comparison with selected breeds often needs to be supported by governmental incentives which sometimes barely cover production costs.

These products’ distinctive features and enhanced quality may be highly valued by consumers. Such an increased popularity of breed-linked products in turn favours investments in local farmers, who act as the main preservatory agents of domestic poultry breed biodiversity [139].

As suggested in González Ariza et al. [140], redundant variables need to be discarded prior to statistical analyses, leading to CHAID decision tree building before the splitting and pruning stages. In line with this premise, on the one hand, L* meat 72 h post-mortem, a* meat 72 h post-mortem, b* meat 72 h post-mortem, and pH 72 h post-mortem were deemed redundant variables and were discarded from further analyses. The basis for these redundancies may rely on the fact that measurements taken at 72 h post-mortem may be poorly representative of carcass quality, especially considering prior sampling. This finding suggests the fact that, in research dealing with the study of products derived from local chicken breeds, quality parameter measurements may not need to be taken exceeding 72 h post-mortem, as this may not report any relevant information which has not been provided by earlier measurements.

On the other hand, multicollinearity problems were also reported for the sex variable. These redundancies may be ascribed to the lack of occurrence of significant differences between females and males in the values reported for the different parameters studied. Indeed, the only empirical differences between sexes concerned the piece or carcass weight and yield variables, and were small but still nonsignificant.

Parallelly, multicollinearity problems were also reported for texture-related traits such as firmness and hardness. Such a strong relationship may also be the source of misconception between the perception of consumers of these two parameters. For example, while increased hardness is always reported as an undesirable feature in meat, an increased firmness may be a sign of a better performance of meat for certain culinary preparations which involve boiling techniques, given meat does not crumble. Specifically, while firmness has been defined as the peak force exerted when a sample was compressed to a depth of 1.5 cm, using a block of wood of identical dimensions to the sample, hardness is defined as the peak force exerted when a metal probe is inserted into the sample to a depth of 1.5 cm [141].

Once redundant variables were discarded, the chewiness and gumminess traits reported the highest value (0.655) in the correlation matrix. Chewiness is the product of gumminess by springiness [142]. Hence, if values of springiness are close to 1, in general, this may be indicative of gumminess and chewiness having similar values, thus being highly correlated texture-related traits as well.

Furthermore, a high positive correlation (0.529) was found between chewiness and muscle fiber diameter. Chewiness is measured performing a sensory evaluation using a simulation of human chewing [143]. The muscle fiber diameter determines the textural characteristics of meat in a determinant moment [144]. A positive correlation between muscle fiber and texture-related characteristics has been reported in previous studies [144], with native chicken breeds presenting high values in the texture profile analysis [145,146].

The shear force/hardness pair of traits reported the second highest positive value in the correlation matrix (0.638). This may derive from the fact that the shear force trait can be defined as the force required to sever a sample of meat [147], while hardness is defined as the peak force required for the first meat compression [148].

In regard to colour-related traits, the b* skin trait highly correlated with L* skin and a* skin (0.459 and 0.561, respectively). Individuals displaying dark skin pigmentation have been reported to present low L*, a*, and b* skin values. In contrast, higher values of L* a* b* indexes were reported for the skin of lighter-coloured birds [149]. Parallelly, when colour coordinates were measured in meat, the highest negative values in the correlation matrix were obtained between L* and a* values. Some authors have proposed that low L* values in meat may most likely be ascribed to high myoglobin concentrations [150]. Additionally, the shift from the glycolytic to oxidative fiber types results in a higher concentration of muscle myoglobin and produces darker meat (higher a* and b* values) in the carcass [151,152].

As depicted in Figure 3, the best discriminating ability was reported for carcass yield. Certain factors such as the genotype and the environment where birds grow may interact, and such an interaction may be the source not only for large differences in the yield that meat cuts and carcass eventually reach, but also for their high variability [153].

Contextually, in local genotypes which are well-adapted to alternative organic or free-range production systems, the development of frequent extensive movements and exercise which compels animals to generate increased kinetic forces is particularly evidenced through the higher development of certain cuts such as the thighs and the drumsticks. In this regard, limb-related cuts are the parts of a bird’s body which most actively participate in the successful development of kinetics, which in turn may explain the increased volumes that they reach [154]. Indeed, even with poultry being considered a species of a ‘white’ meat type, it may not be surprising that increased levels of myoglobin can be found in limb-related areas, which confers them a rather darker aspect derived from this mixed red/white type of fibers, which is even preserved after cooking. Furthermore, as physical exercise increases, fat deposition decreases, which is why lower values of abdominal fat are observed in animals whose life or the greatest part of their life occurs outdoors [155].

Still, yield may be conditioned by other factors such as the age of the individuals or even the breed to which animals belong. In this regard, the yield of the different cuts in the chicken carcass may change along the different stages of growth during the life of the individuals, with breast and thigh yields reaching a greater development than other cuts in older individuals (allometric growth) [156]. This may differ across breeds which have a relatively slow growth, which in turn may be the basis for the great variability found in the slaughter age of the animals.

The pH was a determinant discriminant factor in cuts weighing less than 14.46 g. pH has been related to several meat quality-related attributes including colour, water-holding capacity, tenderness, juiciness, cooking loss, shelf life, and slaughter age [29,58,157]. According to the literature, higher meat pH values are related to rather effective desirable colour retention and moisture absorption properties [158]. Additionally, lower values of pH are related to a rather sour perception of meat flavour by consumers, while higher pH values have been linked to more pleasant, sweeter tastes [159].

In most studies, pH measurement is only taken in the pectoralis major and bicep femoris muscles (breast and thigh muscles). This may be a source of bias given the pH values of thigh muscles are likely higher than those for breast muscles [160]. Muscle exercise increases the number of mitochondria in αW fibers, converting them into αR fibers [161]. This triggers the increase in muscle oxidative capacity to fulfill the needs of the exercise being developed [162]. Additionally, the enhancement of the aerobic catabolism of pyruvate causes a sparing of glycogen, given the oxidative pathway is the most way method to produce energy, which eventually may explain the pH differences across the different meat cuts [163].

A high discriminant potential was reported for the carcass/meat cut weight variables, which were both highly variable traits across breeds. In this context, genotype, environmental factors, and slaughter age have been reported to determine the weight that the different meat cuts reach [154,164]. The weight of the whole carcass and noble cuts, such as breast, thigh, and drumstick, have high economic and environmental importance for poultry meat production since these traits are considered the main production indicators and are the cuts which eventually reach the highest processes due to their appreciation by consumers. However, reaching good production results is conditional to the efficient use of feeds and water [165].

Countries around the world have suffered from the economic impact of the COVID-19 pandemic since 2020. This situation has been exacerbated by the recent Russia–Ukraine war conflict in 2022, as world economies may witness another rise in commodity prices and “supply chain chokeholds” [166]. The world’s largest supplier of wheat is Russia which, together with Ukraine, accounted for about 28% of the sum of global exports during the years 2015–2020 [167].

Within this global framework, autochthonous breeds characterized by specific attributes and features, such as biological breeding, sustainable production system idoneity, and their efficient use of alternative raw materials, may represent a solution to the inflation and economic instability plaguing meat production systems. In this sense, products derived from local genotypes may need to be valued and integrated into the market as quality products, taking advantage of recent trends in consumers, who progressively seek to purchase products which come from less intensive production systems [168]. In this regard, institutional support is necessary to develop investigation studies concerning local breeds, which in turn will act as a protective measurement for these genotypes, considering their ecologic value, and provide oriented market strategies based on a better and conscious valuation of sustainable products [169].

As we progressed in the valuation of the CHAID tree, the third division of the tree subnodes suggested other variables may play an important role in the classification of different meat cuts. Among them, a high discriminant potential was revealed for slaughter age. The basis for such an increase in discriminant potential may stem from the high variability reported for slaughtering age worldwide. Native genotypes are genetically and culturally integrated into the areas from which they come; hence, the determination of age of slaughter mainly affects certain characteristics of the meat that particularly adapt to the local culinary culture of the area in which specific breeds are reared [3,170].

The influence of slaughtering age also explains the high influence of age on the classification of the different pieces in the decision tree, for instance, the high positive correlation with myofiber size. Specifically, larger myofiber diameters and lower myofiber density may translate into larger cut sizes. Indeed, the number of myofibers does not increase after hatching, but meat cut growth is produced by the growth of each myofiber, which may sustain the aforementioned [170].

Simultaneously, as age increases, a decrease in lactate dehydrogenase and phosphofructokinase, which are two glycolytic enzymes, is produced in the chicken muscle. The reduced glycolytic potential produces an increase in pH in older individuals, which is more evident in noble cuts compared to the rest [171,172]. In addition to this, heavier chickens have been reported to present higher plasma glucose and, therefore, are more prone to pre-slaughter stress than lighter ones [173]. This could lead to low muscle glycogen and high pH values at the time of death in older individuals [36,174].

Among other differences between younger and older chickens, the meat of older individuals contains higher myoglobin and collagen proportions [175,176]. It is the variation in concentration in such compounds during the life of the animals which determines the higher or lower repercussion of age on colour, texture, and water-holding capacity-related traits.

Within the gross nutrient cluster, protein presented the highest discriminant potential. Indigenous chickens have been reported to usually present progressively higher protein levels as they age [177]. In this way, the high variability in the slaughter age variable also caused the protein content to vary across the genotypes that were sampled in this study. Nevertheless, the conditioning effects of other factors such as genotype and sex on protein content cannot be discarded, as suggested by the literature [177,178].

Even though the three variables classified within the weight-related traits cluster reported the least relevant discriminating cluster when compared with the rest of the clusters, a relatively high discriminating potential was observed for the cold weight variable when compared to its cluster counterparts. This variable closely relates to the initial carcass/piece weight of individuals. A loss in carcass weight is produced by the action of the cold air in forced circulation when carcasses are conserved into cooling chambers [140].

Last but not the least, L* meat was the only variable to appear as a discriminant criterion in the first three divisions of the decision tree. Chicken meat is translucent; however, when tissues have high pH values, light scattering is weak. This means that the light path through the tissue is relatively long and the selective absorbance of light myoglobin and its derivatives increases. However, in low-pH-value tissues, light scattering is strong, the path of light through the fibers is relatively short, and the selective absorbance of light decreases. Therefore, meat translucency comprehensively and highly influences all meat colourimetry-related parameters [140].

In this context, a large number of colorimeters whose original function is to measure the colour of plastic, metal, or painted surfaces can be found in the market. Such colorimeters are erroneously used in research studies since optical problems derived from the translucency of chicken meat are not taken into account [179]. Instead, the meat colouration of every genotype is matched to the specific requirements of a particular market [180]. Furthermore, the L* meat is also influenced by post-mortem glycolysis. Consequently, chicken nutrition, transport to the slaughterhouse, slaughter, and the refrigeration method used in each culture could contribute to L* meat variation [179].

The present research should be taken into account when deciding which breeds should be used as control and test groups in studies aiming to determine carcass and meat cut quality in chickens. Furthermore, not only the factors that should unavoidably be considered when planning studies are proposed, but also which parameters may hold the greatest capacity to explain intergroup variability, which is eventually the source of significant differences. This enhances the efficiency of the methods used by poultry-related science and maximizes the outcomes derived from future research, which in turn is one of the milestones on which to support autochthonous breed sustainability and preservation.

## 5. Conclusions

Preliminary multicollinearity analyses suggested that meat quality parameters need not be measured after 72 h post-mortem since the information they offer can be supplemented with the rest of the variables collected at the slaughter moment. Small nonsignificant differences between males and females are responsible for the lack of effect of sex on carcass and meat cut quality. Regarding texture-related traits, multicollinearity problems between firmness and hardness may be the source of the misconception between their perception by consumers. On the other hand, gumminess and chewiness variables are highly correlated via their connection to springiness. Native breeds generally present high texture values due to their reduced muscle fiber diameter. The strong relationship between shear force and hardness may derive from conceptual similarities. Individuals displaying dark skin pigmentation present low L*, a*, and b* skin values, as opposed to lighter-coloured skin birds. High myoglobin concentrations in local breeds are responsible for their low L* values. Meat translucency (L*) is also conditioned by slaughtering stress and handling factors, and highly influences all meat colorimetry parameters. Thus, colorimeters may be erroneously used, given that translucency is not considered but directly matched with the specific requirements of particular markets. Slaughtering age, genotype, and environment interaction are the sources of carcass and meat cut yield and weight variability. Higher pH values imply a rather effective desirable colour retention, moisture absorption, and more pleasant sweeter tastes. Reduced glycolytic potential, higher plasma glucose, and proneness to pre-slaughter stress produce an increase in pH and decrease in muscle glycogen in older and heavier individuals, which is more evident in noble cuts. Slaughtering age choice conditions meat characteristics, and in turn is a manner to adapt to the local culinary culture of the area in which specific breeds are reared. Larger cut sizes derive from larger myofiber diameters but lower myofiber density, as meat cut growth is produced by the growth of each myofiber. Moreover, indigenous chickens usually present progressively higher variable protein content with age. The present tool helps to tailor efficient study plans for specific carcass and meat cut quality studies in autochthonous breeds, which in turn may act as strategy reinforcers for local genotype sustainability in the long term.

## Figures and Tables

**Figure 1 animals-12-02702-f001:**
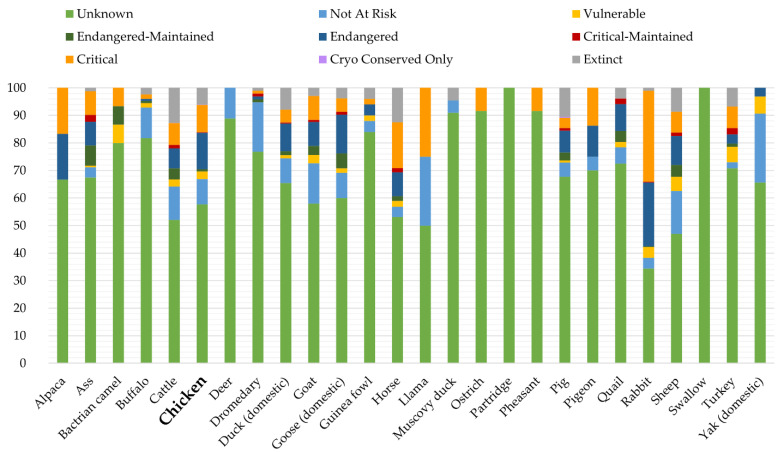
Classification status of breeds across species according to FAO DAD-IS (as of 2021).

**Figure 2 animals-12-02702-f002:**
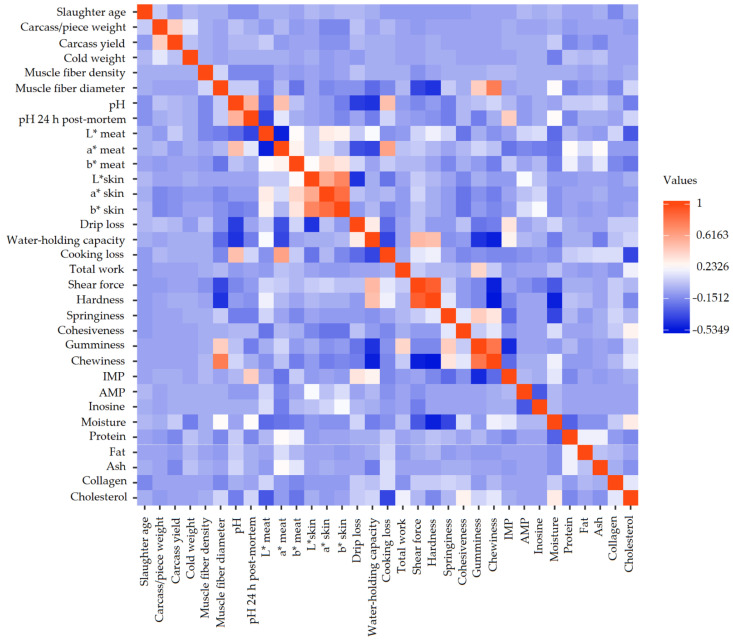
Correlation matrix between the different quality-related traits.

**Figure 3 animals-12-02702-f003:**
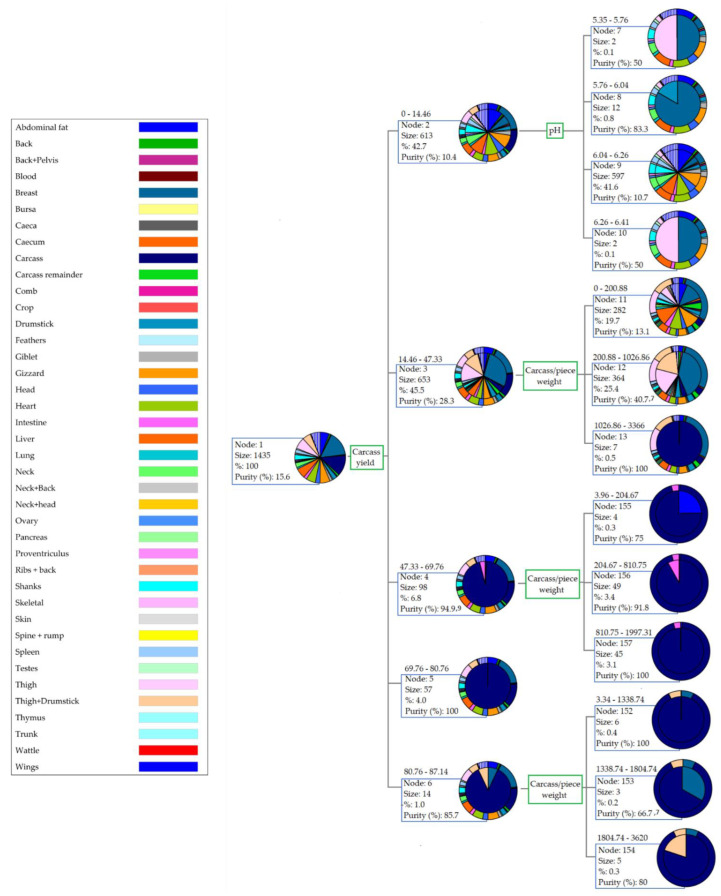
Graphical representation of the first three branches of the CHAID decision tree considering meat cuts as the clustering criterion.

**Table 1 animals-12-02702-t001:** Clusters, units, and references of the traits considered in the studies.

Cluster	Trait	Unit	References
Weight-related traits	Carcass/piece weight	g	[29,30,31,32,33,34,35,36,37,38,39,40,41,42,43,44,45,46,47,48,49,50,51,52,53,54,55,56,57,58,59,60,61,62,63,64,65,66,67,68,69,70,71,72,73,74,75,76,77,78,79,80,81,82,83,84,85,86,87,88,89,90,91,92,93,94,95,96,97,98,99,100,101,102,103,104,105,106,107,108,109,110]
	Carcass yield	%	
	Cold weight	g	
Water-holding capacity	Drip loss	%	[29,31,32,36,38,40,41,42,44,45,52,53,56,57,58,59,61,62,63,65,67,68,70,71,72,73,74,75,76,81,82,84,89,93,103,104,105,108,109,111,112,113,114,115,116,117,118,119,120,121]
	Water-holding capacity	%	
	Cooking loss	%	
Colour-related traits	L* meat		[29,31,32,34,36,38,39,40,41,42,44,45,47,50,52,53,57,58,59,61,62,63,64,65,68,70,71,73,74,75,76,80,82,84,93,98,99,101,103,104,111,112,113,115,117,118,119,121,122]
	a* meat		
	b* meat		
	L* meat 72 h post-mortem		
	a* meat 72 h post-mortem		
	b* meat 72 h post-mortem		
	L* skin		
	a* skin		
	b* skin		
Histological properties	Muscle fiber density	fibers/mm^2^	[35,40,56,65,73,104,105,123]
	Muscle fiber diameter	µm	
Texture-related traits	Firmness	kg s^−1^	[29,32,34,36,38,39,40,41,42,44,45,50,52,53,56,57,58,59,62,63,65,67,70,75,76,81,82,84,93,103,104,105,109,111,112,113,114,115,117,118,119,120,121,122]
	Total work	kg mm	
	Shear force	N	
	Hardness	N	
	Springiness	Mm	
	Cohesiveness	N	
	Gumminess	N	
	Chewiness	kg mm	
pH	pH		[29,30,31,32,33,34,36,38,39,40,41,42,45,47,50,52,53,57,58,59,61,62,63,64,65,67,68,70,71,72,73,74,75,76,82,84,87,89,93,98,99,101,103,104,105,111,112,113,114,115,117,118,119,120,121,122,124,125,126]
	pH 24 h post-mortem		
	pH 72 h post-mortem		
Content of flavour-related nucleotides	IMP	mg/g	[41,48,56,77,109,113,118]
	AMP	mg/100 g	
	Inosine	mg/100 g	
Gross nutrients	Moisture	%	[6,30,33,34,36,38,39,40,41,42,43,47,49,52,55,56,58,59,60,61,64,65,67,68,70,72,75,77,78,80,83,84,87,89,92,93,99,100,102,103,104,105,107,109,111,112,113,114,115,116,119,120,121,124,125,127]
	Protein	%	
	Fat	%	
	Ash	%	
	Collagen	%	
	Cholesterol	mg/100 g	

**Table 2 animals-12-02702-t002:** Multicollinearity analysis of meat and carcass quality-related traits.

Statistics/Traits	VIF ^1^	Tolerance (1 − R^2^),
Chewiness	4.0515	0.2468
Gumminess	3.1989	0.3126
Hardness	2.3258	0.4300
Shear force	2.0546	0.4867
a* meat	1.8862	0.5302
b* skin	1.7745	0.5635
a* skin	1.7044	0.5867
Muscle fiber diameter	1.6223	0.6164
Cooking loss	1.6202	0.6172
L* skin	1.6152	0.6191
L* meat	1.5910	0.6285
Water-holding capacity	1.5580	0.6418
pH	1.4108	0.7088
Drip loss	1.3886	0.7201
pH 24 h post-mortem	1.3486	0.7415
Moisture	1.3462	0.7428
b* meat	1.3408	0.7458
Total work	1.2699	0.7875
IMP	1.2534	0.7978
Springiness	1.2183	0.8208
Cholesterol	1.2101	0.8264
Cohesiveness	1.1135	0.8981
Collagen	1.1130	0.8985
Inosine	1.1058	0.9044
Carcass/piece weight	1.0949	0.9133
Carcass yield	1.0898	0.9176
Protein	1.0761	0.9293
AMP	1.0735	0.9315
Ash	1.0463	0.9558
Muscle fiber density	1.0317	0.9692
Cold carcass weight	1.0275	0.9732
Average age	1.0267	0.9740
Fat	1.0213	0.9792

^1^ Interpretation thumb rule: VIF ≥ 5 (highly correlated); 5 > VIF > 1 (moderately correlated); VIF = 1 (not correlated).

**Table 3 animals-12-02702-t003:** Complexity parameter (Cp) evaluation through the comparison of model-based (resubstitution) statistics and ten-fold cross-validation error rate (risks).

Risk (Cp)	Estimate	Std. Error
Resubstitution error rate	0.604	0.013
Cross-validation error rate	0.622	0.013

## Data Availability

All data stemming from the present research are enclosed in the tables or as Appendix A. Any additional data will be made accessible from the corresponding authors upon reasonable request.

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
