# Peer review of "Data Mining as a Tool to Infer Chicken Carcass and Meat Cut Quality from Autochthonous Genotypes"

_animals, 2022, doi:10.3390/ani12192702_

Round 1
Reviewer 1 Report
GENERAL OVERVIEW: the authors presented a manuscript regarding the inference of the chicken carcass and meat cut quality from autochthonous genotypes using data mining techniques. This is a useful topic due to its importance for the poultry industry to identify the product gaps and manage big data with information about the quality of the product. The authors performed a meta-analytical approach to obtain these data using ninety-one documents. However, in this case, the PRISMA protocol (Preferred Reporting Items for Systematic reviews and Meta-Analyses - Ann Intern Med. 2009;151:264-269) must be used for a reliable meta-analysis. It is mandatory to follow the 27-item checklist and the four-phase flow diagram. Without this, the study will be incomplete and will flaw in reliability. Finally, the figure is impossible to read due to the small font size. I deem it unlikely that all those issues can be solved merely by a few added paragraphs. Thus, I cannot indicate approval of the manuscript in this form. I will avoid the point-by-point review, understanding that a deep correction and a new manuscript must be present in a new submission.
Author Response
Reviewer 1:
GENERAL OVERVIEW: the authors presented a manuscript regarding the inference of the chicken carcass and meat cut quality from autochthonous genotypes using data mining techniques. This is a useful topic due to its importance for the poultry industry to identify the product gaps and manage big data with information about the quality of the product. The authors performed a meta-analytical approach to obtain these data using ninety-one documents. However, in this case, the PRISMA protocol (Preferred Reporting Items for Systematic reviews and Meta-Analyses - Ann Intern Med. 2009;151:264-269) must be used for a reliable meta-analysis. It is mandatory to follow the 27-item checklist and the four-phase flow diagram. Without this, the study will be incomplete and will flaw in reliability. Finally, the figure is impossible to read due to the small font size. I deem it unlikely that all those issues can be solved merely by a few added paragraphs. Thus, I cannot indicate approval of the manuscript in this form. I will avoid the point-by-point review, understanding that a deep correction and a new manuscript must be present in a new submission.
Response: The approach followed in the present systematic review has been reported to be an efficient tool in the scope of animals science specific topics [1-3]. The application of PRISMA guidelines was discarded given PRISMA criteria for systematic reviews was developed in the scope of healthcare research, hence, it does not fit the diverse range and nature of documents in which the information in regards local breeds is made available to the public [4]. This has been supported by studies such as that by Tam et al. [5], who suggested that the adherence level of certain journals to the PRISMA statement does not significantly vary whether they endorse or recommend such a guideline. Furthermore, other authors like Haddaway et al. [6], would report the limited applicability of PRISMA guidelines for reviews in conservation and environmental management.
- González Ariza, A.; Arando Arbulu, A.; Navas González, F.J.; Nogales Baena, S.; Delgado Bermejo, J.V.; Camacho Vallejo, M.E. The Study of Growth and Performance in Local Chicken Breeds and Varieties: A Review of Methods and Scientific Transference. Animals 2021, 11, 2492.
- Iglesias Pastrana, C.; Navas González, F.J.; Ciani, E.; Barba Capote, C.J.; Delgado Bermejo, J.V. Effect of research impact on emerging camel husbandry, welfare and social-related awareness. Animals 2020, 10, 780.
- McLean, A.K.; Gonzalez, F.J.N. Can scientists influence donkey welfare? Historical perspective and a contemporary view. Journal of Equine Veterinary Science 2018, 65, 25-32.
- Page, M.J.; Moher, D.; McKenzie, J.E. Introduction to PRISMA 2020 and implications for research synthesis methodologists. Research Synthesis Methods 2022, 13, 156-163, doi:https://doi.org/10.1002/jrsm.1535.
- Tam, W.W.; Lo, K.K.; Khalechelvam, P. Endorsement of PRISMA statement and quality of systematic reviews and meta-analyses published in nursing journals: a cross-sectional study. BMJ Open 2017, 7, e013905, doi:10.1136/bmjopen-2016-013905.
- Haddaway, N.R.; Macura, B.; Whaley, P.; Pullin, A.S. ROSES RepOrting standards for Systematic Evidence Syntheses: pro forma, flow-diagram and descriptive summary of the plan and conduct of environmental systematic reviews and systematic maps. Environmental Evidence 2018, 7, 1-8.
Reviewer 2 Report
Dear authors,
I have completed the review of the article entitled "Data mining as a tool to infer chicken carcass and meat cut quality from autochthonous genotypes". This study aimed to cluster the carcass and meat quality characteristics of native breeds through data mining and seems to have achieved its purpose. However, the only problem is whether this result can be generalized to commercial genotypes. Because there are many differences between native breeds and commercials due to selection for rapid growth, I think this generalization is not very accurate and further studies are needed for commercials. Apart from these, the paper is clear and well written. The results and discussion are coherent, and the conclusion is appropriately written.

Author Response
Reviewer 2:
I have completed the review of the article entitled "Data mining as a tool to infer chicken carcass and meat cut quality from autochthonous genotypes". This study aimed to cluster the carcass and meat quality characteristics of native breeds through data mining and seems to have achieved its purpose.
Response: We thank the reviewer for his/her kind comment.
However, the only problem is whether this result can be generalized to commercial genotypes. Because there are many differences between native breeds and commercials due to selection for rapid growth, I think this generalization is not very accurate and further studies are needed for commercials.
Response: We agree with the reviewer’s comments. In this study, our objective is not to do research in commercial lines, but to focus on autochthonous genotypes. For this reason, we have removed any sentences in the manuscript that could give lead to confusion.
Apart from these, the paper is clear and well written. The results and discussion are coherent, and the conclusion is appropriately written.
Response: We thank again the reviewer for his/her kind comment.

Reviewer 3 Report
It is undoubtedly an interesting and complete study, taking into account many variables and in a fairly considerable time range. The study is well conducted. I personally have no further comments. Perhaps the only comment is in relation to the databases used, because the authors did not take into consideration the Web of Science, being one of the most reliable databases, since they include papers published with rigorous arbitration. The other comment is in relation to the conclusions, which seem more like a summary of results, and it is easy to get lost in it.
Author Response
Reviewer 3:
It is undoubtedly an interesting and complete study, taking into account many variables and in a fairly considerable time range. The study is well conducted. I personally have no further comments. ´
Response: We thank the reviewer for his/her kind comment.
Perhaps the only comment is in relation to the databases used, because the authors did not take into consideration the Web of Science, being one of the most reliable databases, since they include papers published with rigorous arbitration.
Response: The Web of Science database does not include tools that enable data extraction for analysis. The manuscript text has been modified to explain this problem.
The other comment is in relation to the conclusions, which seem more like a summary of results, and it is easy to get lost in it.
Response: the conclusions section has been modified to facilitate reader comprehension.
Round 2
Reviewer 1 Report
Thank you for reply my questions. However, unfortunately, I did not find consistent reasons to change my prior suggestion. The PRISMA protocol is used worldwide for health studies and other topics of interest, such as the Agrarian Sciences, Environment and Land survey. This protocol has the main objective of organising the information, reducing bias and improving the reliability of the final results. Based on this, I cannot suggest the approval of this manuscript as presented. The authors made a great effort to gather data, and the issue topic is handy. Despite the criticism of the method, this is important to register.